# TREM2 Expression and Amyloid-Beta Phagocytosis in Alzheimer’s Disease

**DOI:** 10.3390/ijms24108626

**Published:** 2023-05-11

**Authors:** Francesca La Rosa, Simone Agostini, Federica Piancone, Ivana Marventano, Ambra Hernis, Chiara Fenoglio, Daniela Galimberti, Elio Scarpini, Marina Saresella, Mario Clerici

**Affiliations:** 1IRCCS Fondazione Don Carlo Gnocchi, 20147 Milan, Italy; sagostini@dongnocchi.it (S.A.); fpiancone@dongnocchi.it (F.P.); imarventano@dongnocchi.it (I.M.); ahernis@dongnocchi.it (A.H.); msaresella@dongnocchi.it (M.S.); mario.clerici@unimi.it (M.C.); 2Department of Pathophysiology and Transplantation, University of Milan, 20100 Milan, Italy; 3Fondazione Ca’ Granda, IRCCS Ospedale Maggiore Policlinico, 20122 Milan, Italy; 4Department of Biomedical, Surgical and Dental Sciences, University of Milan, 20100 Milan, Italy

**Keywords:** Alzheimer’s disease, Ab-phagocytosis, peripheral monocytes, research biomarker, TREM2

## Abstract

Alzheimer’s Disease is the most common form of dementia; its key pathological findings include the deposition of extracellular-neurotoxic-plaques composed of amyloid-beta (Ab). AD-pathogenesis involves mechanisms that operate outside the brain, and new researches indicate that peripheral inflammation is an early event in the disease. Herein, we focus on a receptor known as triggering-receptor-expressed-on-myeloid-cells2 (TREM2), which promotes the optimal immune cells function required to attenuate AD-progression and is, therefore, a potential target as peripheral diagnostic and prognostic-biomarker for Alzheimer’s Disease. The objective of this exploratory study was to analyze: (1) soluble-TREM2 (sTREM2) plasma and cerebrospinal fluid concentration, (2) TREM2-mRNA, (3) the percentage of TREM2-expressing monocytes, and (4) the concentration of miR-146a-5p and miR-34a-5p suspected to influence TREM2 transcription. Experiments were performed on PBMC collected by 15AD patients and 12age-matched healthy controls that were unstimulated or treated in inflammatory (LPS) conditions and Ab_42_ for 24 h; Aβ_42_-phagocytosis was also analyzed by AMNIS FlowSight. Results although preliminary, due to limitations by the small sample-size, showed that in AD compared to HC: TREM2 expressing monocytes were reduced, plasma sTREM2 concentration and TREM2-mRNA were significantly upregulated and Ab_42_-phagocytosis was diminished (for all *p* < 0.05). miR-34a-5p expression was reduced (*p* = 0.02) as well in PBMC of AD, and miR-146 was only observed in AD cells (*p* = 0.0001).

## 1. Introduction

The mechanisms by which innate immune responses contribute to neuroinflammation and neurodegeneration are only partially understood. The triggering receptor expressed on myeloid cells-2 (TREM2) protein is a transmembrane protein belonging to the TREM family and is an important innate immune receptor in the brain. TREM2 is primarily expressed by microglia, myeloid cells that in the central nervous system (CNS) are involved in immune surveillance, cell-cell interactions, control of latent inflammatory reactions, and tissue debris clearance [1,2,3,4,5]. Thus, TREM2 plays an important role in microglial phagocytosis of apoptotic neurons, damaged myelin, and amyloid plaques [6,7,8,9,10,11,12,13]. Furthermore, TREM2 regulates microglial biosynthetic metabolism [14], proliferation, and survival [15], cytokine release [16], and their accumulation around plaques. SNPs in the TREM2 gene modulate protein expression [17,18] and function, possibly resulting in an inactive receptor [17]. Such single nucleotide polymorphisms (SNPs) were identified as risk factors for Alzheimer’s disease (AD), a central nervous system disease that leads to dementia [19] and it is characterized by progressive cognitive dysfunction, memory loss, and neuroinflammation [20,21,22,23,24,25,26,27]. Recently, two independent studies reported that a heterozygous rare variant in TREM2 p.R47H is significantly associated with AD [28,29] and multiple variants in the same gene increase AD risk [30,31,32]. 

According to amyloid cascade theory [33] the presence of extracellular plaques of insoluble beta-amyloid peptide (Aβ) and neurofibrillary tangles (NFT) of P-tau in neuronal cytoplasm is the hallmark of AD [34]. The cerebral accumulation both extra- and intraneuronally of Aβ peptide, resulting from the imbalance between production and clearance of this protein, is the main event causing the disease [34]. In the AD brain, Aβ clearance can be mediated by drainage to perivascular spaces or by Aβ uptake; it is processing by different cell types [35,36]. When activated by ligands, such as Aβ, TREM2 induces an innate immune response, which includes phagocytosis, chemotaxis, and transcriptional changes [37]. The impact of TREM2 on plaque accumulation in amyloid pathology was examined in TREM2-deficient AD murine models, but results were conflicting [6,7,8,9,10,11,12,13,14,15,16,17]. Interestingly, a later study suggested disease progression-dependent effects of TREM2 on amyloid pathology. TREM2 deficiency ameliorates amyloid pathology in the early phases of the disease, but exacerbates it late in the disease process [38]. Notably, while resident microglia plays a key role in Ab-clearance in the brain, approximately 40–60% of brain-generated Aβ is estimated to diffuse into the blood and being cleared in the periphery, implying that peripheral mechanisms play an essential role as well in Aβ clearing [37,39].

TREM2 undergoes regulated proteolytic cleavage in the H157-S158 peptide bond by two proteins, ADAM10 and ADAM17, resulting in the generation of a soluble form of the protein (sTREM2) [40,41,42]. sTREM2 can be measured in plasma and in CSF [43,44,45]. sTREM2 concentration changes dynamically during the progression of AD, peaking at the early symptomatic stages of the disease [43,44,45,46]. sTREM2 CSF concentrations were reported to correlate with neuronal injury markers, including the CSF concentration of total tau and phospho-tau [47]. Notably, TREM2 mRNA detected in peripheral monocytes and sTREM2 serum concentration were shown to directly correlate with clinical parameters for AD diagnosis [48,49]. Therefore, in this study, we investigated peripheral TREM2 expression and the concentration of 4 miRNAs (miR-146a-5p, miR-125b, miR-9-3p, miR-34a-5p), which are differentially expressed in AD and are suspected to influence the rate of TREM2 mRNA transcription [50], in AD patients and HC. Results herein show that sTREM2 plasma concentration and mRNA are increased whereas TREM2-expressing monocytes are reduced in AD. This was correlated with an impairment of Aβ_42_-phagocytosis. Further data indicating a direct correlation between miR-34a-5p concentration and TREM2-mRNA and sTREM2 plasma concentration suggest a possible regulatory role for this miRNA on TREM2 biosynthesis.

## 2. Results

### 2.1. Patients and Control

The demographic and clinical characteristics of the individuals enrolled in the study are summarized in Table 1. No differences were observed in gender, age, or years of education or genotype distribution of ApoΕ 4 carrier. The allele frequency of ApoE 4 that was found in AD patients and in the control population (HC) was 20%; in detail: 4 of the 15 patients carried the ApoE 4 allele (E4^+^): 2 were E4^+^/E4^+^ homozygote and 2 were E4^+^/E3^+^ heterozygote. Among *Apo* E4^−^/E4^−^ subjects, all 12 patients carrier the E3^+^/E3^+^ genotype. In HC: 3 of 12 subjects included in the study were E4^+^/E3^+^ heterozygote and 1 was E2^+^/E4^+^; 8 were E3^+^/E3^+.^ As per inclusion criteria, global cognitive levels (MMSE) were significantly reduced in AD patients (median 21 ± 3.9) compared to HC (>28) (*p* < 0.05).

### 2.2. sTREM2 in CSF and Plasma

Soluble TREM2 (sTREM2) was quantified in the plasma of HC and AD patients and in CSF of AD alone. Results obtained in plasma showed that sTREM2 was significantly increased in AD (5189 ± 768 pg/mL) compared to HC (3681± 463 pg/mL) (*p* = 0.01) (Figure 1A). Within the group of AD patients, sTREM2 concentration was significantly increased in CSF (7183 ± 287 pg/mL) compared to plasma (*p* = 0.01) (Figure 1B).

### 2.3. TREM2 mRNA in Monocytes of AD and HC

A two-fold increase in TREM2 mRNA expression was observed in unstimulated (medium) monocytes of AD (median = 2488 copies/ng) compared to HC (median = 1230 copies/ng) (Figure 2); this difference approached but did not reach statistical significance. TREM2 was also quantified in CSF of AD patients, but mRNA concentration in CSF was below the limit of detection. Similar results were obtained in LPS +Ab_42_ stimulated cells.

### 2.4. TREM2 Expressing Monocytes in AD and HC

The percentage of TREM2-expressing cells was examined by flow cytometry in a subset of subjects, selected according to the availability of enough cells to run the experiment. Thus, CD14^+^ immune cells of 10 AD patients and 10 control were either unstimulated or stimulated with LPS and Aβ_42_ for 24 h. In unstimulated conditions (medium), the percentage of TREM2 expressing monocytes was increased although not significantly in HC (median 3.2%) compared to AD patients (median 0.9%). Interestingly, TREM-2 expression on monocytes of HC was significantly reduced (*p* < 0.05) when cells were stimulated with LPS + Aβ_42_, suggesting a negative effect of inflammatory stimulation on TREM2 expression (Figure 3).

### 2.5. Aβ_42_-Phagocytosis by Peripheral Monocytes of AD Patients and HC

Aβ_42_ phagocytosis was examined by evaluating the percentage of monocytes showing internalized Aβ_42_FAM-(labeled with FITC-conjugated) in Aβ_42_-FAM alone or in Aβ_42_-FAM+LPS- stimulated cell cultures. Results showed that the percentage of Aβ_42_-FAM-positive cells was significantly reduced in AD patients (18%) compared to HC (28%) (*p* < 0.05). Notably, the percentage of monocytes with internalized Aβ_42_ was significantly reduced in Aβ_42_-FAM + LPS (0.5%) compared to Aβ_42_-FAM (21%) stimulated cells of HC (*p* = 0.01). Taken together with the reduction of TREM2 expression seen when LPS was added to cell cultures (see above), this result confirms that Aβ_42_ phagocytosis is reduced, at least in vitro, in presence of an inflammatory milieu (Figure 4). Appendix A: Figure 3 and Figure 4 were merged in the same panel.

### 2.6. miR-34a-5p and miR-146a-5p Are Differently Expressed in AD Patients and HC

miR-34a-5p expression was decreased in unstimulated (medium) AD monocytes (median = 89 copies/pg) compared to what was seen in HC (median = 989 copies/ng) (Figure 5A) and was not significantly modified by LPS + Aβ_42_ stimulation in either group (Figure 5A). miR-146a-5p could only be detected in AD monocytes, both in medium (median = 37,181; copies/ng) and post LPS + Aβ_42_ stimulation (median = 21,100; copies/ng) (Figure 5B). miR-34a-5p and miR-146a-5p expression was also investigated in CSF of AD patients, but their concentration in CSF was below the limit of detection. Finally, the concentration of miR-125b and miR-9-3p was below the limit of detection both in AD and HC in all experimental conditions. Appendix A: Figure 2 and Figure 5 were merged in the same panel.

## 3. Discussion

Alzheimer’s disease (AD) is the most common type of dementia and imposes substantial economic and social burdens [51]. Biomarkers are crucial for the accurate and early identification of AD and are a prerequisite for the effective management of the disease.

Immunity and inflammation are essential processes at play throughout the whole AD process, and the related biomarkers could be part of the diagnosis.

Since the establishment of the common protocol for the AD, cerebrospinal fluid tests and positron emission tomography examinations have become widely accepted. However, problems with invasiveness and high cost limit the application of the above diagnostic methods aimed at the central nervous system. Therefore, different studies suggest a focus on peripheral biomarkers in the diagnosis of AD; also, the structures of biomarkers or a combinations of interacting biomarkers and the methodologies for the detection are important for the exploration of AD.

Triggering receptor expressed on myeloid cells-2 (TREM2) is a receptor in the microglial membrane; it interacts directly with Aβ, which restricts the pathological enhancement of Aβ and tau [52]. TREM2 is expressed on cells of the myeloid lineage, including microglia and monocyte-derived macrophages [18], and plays a key role in hampering neuroinflammation by inhibiting the persistent activation of microglia, promoting phagocytosis, and clearing apoptotic neurons [53]. Because CSF concentration of sTREM2 correlates with CSF levels of total tau and phospho-tau, possibly being a biomarker for neurodegeneration [47,53], we verified whether TREM2 expression on peripheral immune cells and sTREM2 plasma concentration could be used as easily accessible peripheral biomarkers for the onset and/or progression of AD. To this end, we analyzed TREM2 RNA, its expression on monocytes as well as sTREM2 plasma concentration in a cohort of Alzheimer’s patients (AD) comparing results to those obtained in a control population of healthy subjects matched for sex and age. Results herein confirm data obtained previously [54] and suggest that TREM2 expression is altered in AD and helps to define the features of this immune receptor in peripheral blood. We observed that sTREM2 can be measured both in CSF and plasma; CSF concentrations being significantly higher than those seen in plasma. Thus, a higher plasmatic concentration of sTREM2 was found in our well-characterized group of AD patients compared to controls without cognitive decline. It is noteworthy that, while CSF sTREM2 levels have been extensively studied, only a few studies have investigated sTREM2 in plasma, often producing conflicting results [55,56,57,58]. Indeed, existing studies report no significant difference in the plasmatic levels of sTREM2 between AD and HC [59,60]. It has been speculated that changes in sTREM2 concentration in these biofluids are directly correlated with the entity of microglial dysfunction and neuroinflammation in AD [56]. Although it is known that the CSF sTREM2 concentration is elevated in AD [47], it remains unknown how sTREM2 impacts amyloid pathology. Nevertheless, different data suggest that sTREM2 possesses important biological and pathological properties other than acting as a decoy receptor that opposes full-length TREM2 signaling.

Herein we have shown that in AD sTREM2 concentration was increased concomitantly with a decrease of TREM2-expressing monocytes, a reduction of TREM2 level of receptor expression and Aβ_42_-phagocytosis by such monocytes Taken together these results suggest that in vitro Aβ_42-_phagocytosis is impaired in AD as a consequence of the ligation of monocytes-expressed TREM2 by the increased amounts of its soluble, decoy form. 

We also evaluated TREM2 mRNA expression and its possible miRNA targets, including miRNA-34a-5p. The miRNA-34a-mediated down-regulation of TREM2 appears to be recognition feature within the 299 nt TREM2 mRNA 3′-UTR region. Indeed, it is known that epigenetic mechanisms involving miRNA-34a up-regulation and consequent down-regulation of TREM2 expression may drive the progressive extinction of the phagocytic response that in turn contributes to dysfunctional innate-immunity, amyloidogenesis and inflammatory neurodegeneration [61]. Increased TREM2 mRNA expression was described in peripheral blood mononuclear cells from mild cognitive-impaired patients that later converted to AD [62], as well as in blood from subjects with an increased risk to develop dementia [63]. Here, we found a higher TREM 2-mRNA expression in monocytes of AD compared to HC; in the same individuals miRNA-34a-5p expression was reduced. So far, the leading factors that induce TREM2 upregulation in AD-related conditions remain unclear. One possibility is that the components of Aβ plaques synergistically regulate TREM2. This is supported by a study from Neher’s group that TREM2 upregulation is triggered in microglia during their migration to plaques [64]. Another possibility is that TREM2 was induced by multiple factors of neuroinflammation [65]. Although levels of TREM2 are significantly different between unstimulated and stimulated HC, miR-34 is similar. Was reported that whereas TREM2 is downregulated by NF-κB-mediated miRNA-34a, lipopolysaccharides (LPS)-induced pro-inflammatory signaling, as well as pro-inflammatory cytokines. In addition, the TREM2 gene may be a source of epigenetic regulators aimed at self-regulation or modulation of other genes’ expression [66,67,68,69,70,71,72]. Recently, circRNAs have emerged as interesting molecules that deserve to be investigated as epigenetic regulators [73].

Besides miRNA-34a-5p, the only other mRNA that could be measured in our study was miR-146a-5p, a key regulator of the immune response [74] that has been implicated in multiple neuroinflammatory processes, including AD [75]. Thus, studies in murine models of AD and ex vivo results showed that miR-146a expression increases with disease progression and correlates with senile plaque density and synaptic pathology [74,75]. Our results showed that miR-146a-5p could be detected in AD patients alone. This mRNA was shown to reduce TREM2 expression and Aβ-phagocytosis; all these findings were present in our study. These results support the idea that increased miR-146a levels could downregulate the expression of TREM2, leading to reduced Aβ clearance [75]. Moreover, our study results confirmed the idea that increased miR146 expression correlates with reduction of Aβ-phagocytosis in AD patients compared to HC.

Aβ-phagocytosis was down-regulated in LPS +Aβ stimulated monocytes of HC, indicating that in vitro inflammatory conditions down-regulate such process. Previous results indicated that the reduction of TREM2 expression in microglia and macrophages results in a decreased phagocytosis of apoptotic neurons [8,76,77,78], cellular debris [77] and bacteria or bacterial products [79,80,81,82]; notably, increasing TREM2 expression was associated with a more efficient phagocytosis of these substrates [73,74,75,76,77,78,79,80,81,82,83,84]. TREM2 mRNA expression is modulated by inflammation. Thus, in vitro results showed that an anti-inflammatory milieu results in an upregulation of TREM2 expression [85], while pro-inflammatory proteins, such as TNFα, IL1β or lipopolysaccharide (LPS) decrease TREM2 expression [86,87,88]. Opposite results were described in vivo, as an increased expression of TREM2 was observed both in microglial cells and in hippocampus of post-mortem brain of AD patients [89,90,91] and in mouse models of amyloid and tau pathology [85,92,93,94,95], possibly being associated with the recruitment of microglia into amyloid plaques [94,95,96,97]. Acute inflammatory conditions, as those mimicked by in vitro studies, could thus reduce TREM2 expression, while the chronic inflammation observed in pathological conditions, such as AD, would results in increased TREM2 expression, possibly in the attempt to augment phagocytosis.

To summarize results presented herein, in AD patients compared to the HC: (1) the expression of TREM2 is increased, that of its negative feedback inhibitor mir34a-5p is reduced; (2) the one of miR-146a, a TREM2 down-regulator, is increased; (3) TREM2 expression on monocyte is reduced whereas sTREM2 plasma concentration is increased, and (4) Aβ-phagocytosis is decreased. Taken together these results support an important role for miRNA-34a-5p and miR-146a in regulating TREM2 expression. These data also allow the speculation that in AD a significant amount of the TREM2 that is translated into protein undergoes cleavage into sTREM2; it is released into biological fluids, not being available as a cellular receptor to facilitate Aβ-phagocytosis.

## 4. Materials and Methods

### 4.1. Patients and Controls

Fifteen patients who fulfilled the inclusion criteria for the clinical diagnosis of AD were enrolled by the Neurology Clinic of Fondazione Cà Granda, IRCCS, Ospedale Maggiore Policlinico in Milan, Italy. All patients underwent complete medical and neurological evaluation, laboratory analysis, CT scan or MRI, as well as EEG, SPET scan, and CSF examination to exclude reversible causes of dementia. CSF biomarkers Aβ, total tau (total-tau), and tau phosphorylated at position 181 (p-tau) were analyzed; cut-off thresholds of normality were: Aβ ≥ 600 pg/mL; tau ≤ 500 pg/mL for individuals older than 70 and ≤450 pg/mL for individuals aged between 50 and 70 years; P-tau ≤ 61 pg/mL [98]. The clinical diagnosis of AD was performed according to the NINCDS-ADRDA work group criteria [99] and the DMS IV–R [100]. Neuropsychological evaluation and psychometric assessment were performed with a Neuropsychological Battery that included: MiniMental State Examination (MMSE), Digit Span Forward and Backward, Logical Memory and Paired Associated Words Tests, Token Test, supra Span Corsi Block Tapping Test, Verbal Fluency Tasks, Raven Colored Matrices, the Rey Complex Figure, and the Clinical Dementia Rating Scale (CDR) [101]. After the diagnosis, all AD patients were enrolled in a cognitive-functional rehabilitation program.

Twelve sex and age-matched healthy controls (HC) were enrolled as well in the study; these individuals were volunteers without a family history of dementia or evidence of acute or chronic neurologic diseases at the time of enrollment and were selected according to the SENIEUR protocol for immuno-gerontological studies of European Community’s Control Action Programme on Aging [102]. The cognitive status of HC was assessed by MMSE (score for inclusion as normal control subjects ≥ 30). ApoE genotyping was determined in all individuals by allelic discrimination [103]. The study conformed to the ethical principles of the Helsinki Declaration and was approved by the Institutional Review Board of the Fondazione Cà Granda, IRCCS Ospedale Maggiore Policlinico (Milan, Italy). All patients (or their legal guardians) and controls gave their written informed consent before entering the study.

### 4.2. Blood Sample Collection, Cell Separation, and Cell Culture

Whole blood and plasma were collected in vacutainer tubes containing ethylenediaminetetraacetic acid (EDTA) (Becton Dickinson & Co., Rutherford, NJ, USA). Peripheral blood mononuclear cells (PBMC) were separated on lymphocyte separation medium (Cedarlane, Hornby, Ontario, CA, USA) and washed twice in PBS at 1500 RPM for 10 min; viable leukocytes were determined using a TC20 Automated Cell Counter (Bio-Rad, Hercules, CA, USA). PBMC were seeded at a density of 4 × 10^6^/mL on plastic plates-6-wells, and were cultured with RPMI 1640 supplemented with 10% human serum, 2 mM L-glutamine, and 1% penicillin (Invitrogen, Ltd., Paisley, UK) overnight at 37 °C with 5% CO_2_ in a humidified atmosphere to allow monocytes to adhere to the plate. The following day, the medium was changed and monocytes were incubated with Lypopolisacaride (LPS) (1 μg/mL) (Sigma-Aldrich, St. Louis, MO, USA) and Aβ_42_ (2.5 μM) (Phoenix Pharmaceuticals, Burlingame, CA, USA) (Appendix A), or cultured with Alexa Fluor-488 (FAM)-labeled Aβ42 (AS-60479-01) DBA (Segrate, Italy) for 24 h at 37 °C in a humidified 5% CO_2_ atmosphere. One day later supernatants were collected and centrifuged; adherent cells were treated with Accutase (CliniSciences, Nanterre, France) and their viability was determined using a TC20 Automated Cell Counter (Bio-Rad, Hercules, CA, USA).

### 4.3. CSF Collection and Aβ and Tau Determination

CSF samples were collected by lumbar puncture (LP) in the L3/L4 or L4/L5 interspace. The LP was done between 8 and 10 a.m. after one-night fasting. CSF samples were then centrifuged (1500 rpm for 10 min at 4 °C). Supernatants were aliquoted in polypropylene tubes and stored at −80 °C. CSF cell counts, glucose, and proteins were measured; Aβ, total-tau, and p-tau were evaluated using commercially available ELISA kits (Fujirebio, Ghent, Belgium).

### 4.4. Total mRNA Extraction

Total RNA (mRNA and miRNA) was extracted from unstimulated or stimulated (see above) monocytes of all individuals, as well as from CSF of AD patients, using a column-based kit (for blood: miRNeasy Mini Kit, Qiagen GmbH, Hilden, Germany; for CSF: miRNeasy serum/plasma kit, Qiagen, Hilden, Germany) according to the manufacturer’s protocol. RNA concentration was determined by a spectrophotometer (Nanoview plus^TM^, GE Healthcare, Little Chalfont, UK). Purity was determined as the 260/280 nm OD ratio, with the expected values between 1.8 and 2.0. RNA was treated with TURBO DNA-free DNAse (Ambion Inc., Austin, TX, USA). RNA was quantified by Qubit (ThermoFisher Scientific, Waltam, MA, USA). For miRNA experiments, an equal concentration of extracted miRNAs was retro-transcribed in cDNA (miRCURY LNA RT kit, Qiagen, Hilden, Germany) for all samples.

### 4.5. TREM2 mRNA and miRNAs Detection by Droplet Digital PCR (ddPCR)

*TREM2* mRNA and miR-34a-5p, miR-9-3p, miR-125b and miR-146a-5p quantitation were performed by droplet digital PCR (ddPCR QX200, Bio-Rad, Hercules, CA, USA). For *TREM2* gene quantitation, 5 µL of diluted RNA (1:100) was mixed with TREM2 specific primers (Qiagen, Hilden, Germany) and One-Step RT ddPCR Mastermix (Bio-Rad, Hercules, CA, USA), whereas for miRNAs, 3 µL of diluted cDNA (1:10,000 for miR-34a-5p, 1:1000 for miR-146a-5p, 1:10,000 for miR-125b, and 1:10,000 for miR-9-3p) where mixed with LNA^TM^ specific primers (Qiagen, Hilden, Germany) and ddPCR EvaGreen SuperMix (Bio-Rad, Hercules, CA, USA). In both cases, the mix was emulsified with droplet generator oil (Bio-Rad, Hercules, CA, USA) using a QX200 droplet generator, according to the manufacturer’s instructions. Droplets were transferred to a 96-well reaction plate and heat-sealed with a pierceable sealing foil sheet (PX1, PCR plate sealer, Bio-Rad, Hercules, CA, USA). PCR amplification was performed in a sealed 96-well plate using a T100 thermal cycler (Bio-Rad, Hercules, CA, USA). For mRNA, the cycles were: 3 min at 25 °C, 60 min at 50 °C, 10 min at 95 °C, 45 cycles at 95 °C for 15 s and at 60 s at 60 °C, then 10 min at 98 °C and finally hold at least for 30 min at 4 °C. For miRNAs, the cycles were: 10 min at 95 °C, 40 cycles at 94 °C for 30 s, and at 58 °C for 60 s, followed by 10 min at 98 °C and hold at 4 °C. The 96-well plates were then transferred to a QX200 droplet reader (Bio-Rad, Hercules, CA, USA). Each well was queried for fluorescence to determine the number of positive events (droplets); results were displayed as dot plots. The miRNA and mRNA concentration was expressed as copies/ng of extracted RNA.

### 4.6. Phagocytosis Assay by AMNIS FlowSight Imaging Analysis

For the Aβ-FAM-phagocytosis assay, pellets collected from adherent cells were fixed with 0.1% paraformaldehyde (PFA) for 10 min, washed, and resuspended in 50 μL PBS. Analyses were performed by Amnis FlowSight Imaging Flow Cytometer (Luminex Corporation, Austin, TX, USA) an imaging flow cytometer equipped with two lasers operating at 488 and 642 nm, two camera, and twelve standard detection channels that merge flow cytometry and high-resolution microscopy. The machine simultaneously produces side scatter (darkfield) images, one or two transmitted light (brightfield) images, and up to ten channels of fluorescence imagery of every cell. FlowSight using the Inspire^TM^ system acquires 2000 cells/s and operates with a 1 μm pixel size (~20× magnification) allowing visualization of fluorescence from the membrane, cytoplasm, or nucleus. The IDEAS image analysis software allows the quantification of cellular morphology and fluorescence at different cellular localizations by defining specific cellular regions (masks) and mathematical expressions that uses image pixel data or masks (features). Phagocytosis was evaluated by analyzing the internalization feature utilizing a mask representing the whole cell, defined by the brightfield (BF) image, and an internal mask defined by eroding the whole cell mask to eliminate the fluorescent signal coming from the Aβ_42_-FAM attached to the cell surface, thus measuring only the internalized part. The internalization feature was first used to calculate the ratio of the intensity of FAM (Aβ_42_ signal) inside the cell/the total FAM intensity outside the cell. Higher internalization scores (IS) indicate a greater concentration of Aβ_42_ FAM inside the cell.

### 4.7. Detection of sTREM2 Protein by ELISA

TREM-2 concentration was analyzed by sandwich immunoassays according to the manufacturer’s instructions (Abcam, Cambridge, UK) (cod. ab224881) in plasma of AD patients and HC as well as in CSF of AD patients. A plate reader (Sunrise, Tecan, Mannedorf, Switzerland) was used and optical densities (OD) were determined at 450/620 nm. Sensitivity (S) and Assay Range (AR) were as follows: 10.5 pg/mL (S), 78.1–5000 pg/mL (AR).

### 4.8. Immunofluorescent Staining, and TREM2 Expressing Percentage Monocyte Analysis by Flow Cytometry

TREM2 protein expression was measured using an extracellular staining assay. Briefly, unstimulated and stimulated monocytes were washed in PBS, and stained with anti-CD14 PC-7 (IgG2a Mouse, clone: RMO52) (Beckman-Coulter, Rome, Italy); and anti-TREM2 alexa flour 488 conjugated (IgG2a Mouse, clone: 237920) (Bio-Techne, Milan, Italy) mAb for 30 min at 4 °C in the dark. Cells were then washed and fixed using a FIX-kit (Invitrogen-Caltag Lab, Carlsbad, CA, USA) for 30 min at 4 °C; then cells were washed and resuspended in PBS. For the analysis, 20.000 events were acquired and gated on Forward (FSC) and Side scatter (SSC) properties; the monocyte gate was designed on CD14^+^/SSC. Analyses were performed using a Beckman-Coulter GALLIOS flow cytometer equipped with a 22 mW Blue Solid State Diode laser operating at 488 nm and with a 25 mW Red Solid State Diode laser operating at 638 nm and interfaced with Kaluza analysis software. Flow cytometry compensation was performed using the fluorescence minus one (FMO) approach. Briefly, all antibody conjugates in the experiment are included except the one that is controlled for. The FMO measures the spread of fluorescence from the other staining parameters into the channel of interest, determining the threshold for positive staining.

### 4.9. Statistical Analysis

Data analysis was performed using GraphPad Software Inc. (San Diego, CA, USA). For sTREM2 data, normally distributed, Student’s *t*-test was used. For mRNA and miRNAs expression as well as for Aβ-phagocytosis and TREM2 expressing percentage Monocyte analysis, not normally distributed, Mann Whitney test was used to compare the expression value in AD and HC and Wilcoxon matched-pair test for comparison before and after stimuli. For ddPCR analysis, the QuantaSoft software version 1.7.4.0917 (Bio-Rad, Hercules, CA, USA) was used to quantify mRNA and miRNA copies. Thresholds were determined manually for each experiment, according to the negative controls, which included a no template control. droplet positivity was determined by fluorescence intensity, only droplets above a minimum amplitude threshold were counted as positive. Positive controls as well as negative controls were included in each experiment. Samples that resulted in less than two positive droplet controls are considered negative [104]. All the normally-distributed data are summarized as mean ± standard error of the mean (SEM), whereas non-normally distributed data are summarized as median and Interquartile (IQR) (25° and 75° percentile). *p* values of less than 0.05 were considered statistically significant.

## 5. Conclusions

We are aware that there are limitations to this study: thus, the number of the study participants is limited, and these results need to be further investigated in large cohorts of patients. Secondly, we could not analyze these parameters in the CSF of HC. Finally, in vitro experimentation: cell cultures cannot mimic human. These limitations notwithstanding, our data reinforce the notion that TREM2 plays an important role in AD, shed some light on the mechanisms responsible for its regulation, and suggest that TREM2 measurement in peripheral blood could be a useful biomarker for AD diagnosis and prognosis. As future prospects, we hope to also investigate *TREM2* gene-related mutations on peripheral immune cells and its implications on the course of AD.

## Figures and Tables

**Figure 1 ijms-24-08626-f001:**
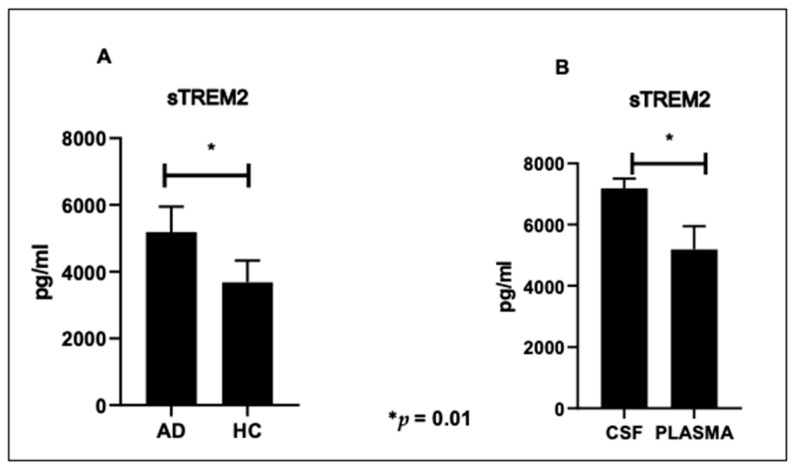
sTREM2 concentration in plasma of 15 Alzheimer’s Disease patients (AD) and 15 controls (HC) (**A**). sTREM2 levels in plasma and CSF of AD patients (**B**). Values are expressed as mean ± standard error of the mean (SEM), *p* values of less than 0.05 were considered significant (* *p* = 0.01).

**Figure 2 ijms-24-08626-f002:**
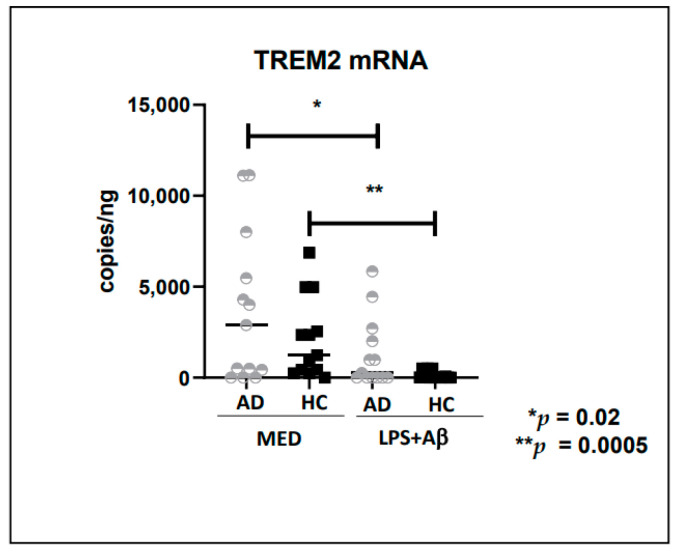
Expression of TREM2 mRNA in unstimulated (MED) or LPS + Aβ_42_ stimulated- PBMC of 13 Alzheimer’s Disease patients (AD) and 12 healthy controls (HC). The Mann Whitney test was used to compare the mRNA or miRNA expression; the Wilcoxon matched-pair test was used to compare results obtained in different cultural conditions. For ddPCR analysis, the QuantaSoft software version 1.7.4.0917 (Bio-Rad, Hercules, CA, USA) was used to quantify mRNA and miRNAs copies. Data are summarized as median and Interquartile (IQR) (25° and 75° percentile. *p* values of less than 0.05 were considered significant (* *p* = 0.02; ** *p* = 0.005).

**Figure 3 ijms-24-08626-f003:**
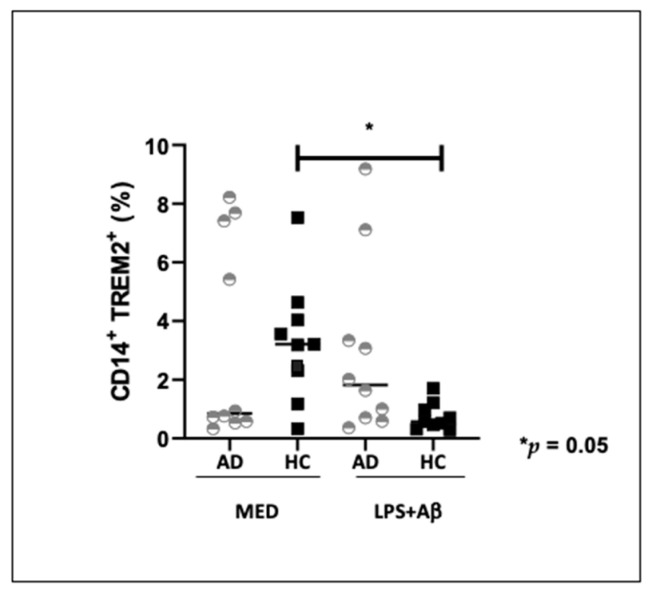
TREM2-expressing CD14^+^ monocytes in 10 Alzheimer’s Disease patients (AD) and 10 healthy controls (HC); results obtained when cells were cultured in medium alone (MED) or upon LPS and Aβ stimulation are shown The Mann Whitney test was used to compare the percentage of TREM2-expressing CD14^+^ monocyte in AD and HC, and Wilcoxon matched-pair test for comparison before and after LPS stimulation. Data are summarized as median and Interquartile (IQR) (25° and 75° percentile). *p* values of less than 0.05 were considered significant (* *p* = 0.05).

**Figure 4 ijms-24-08626-f004:**
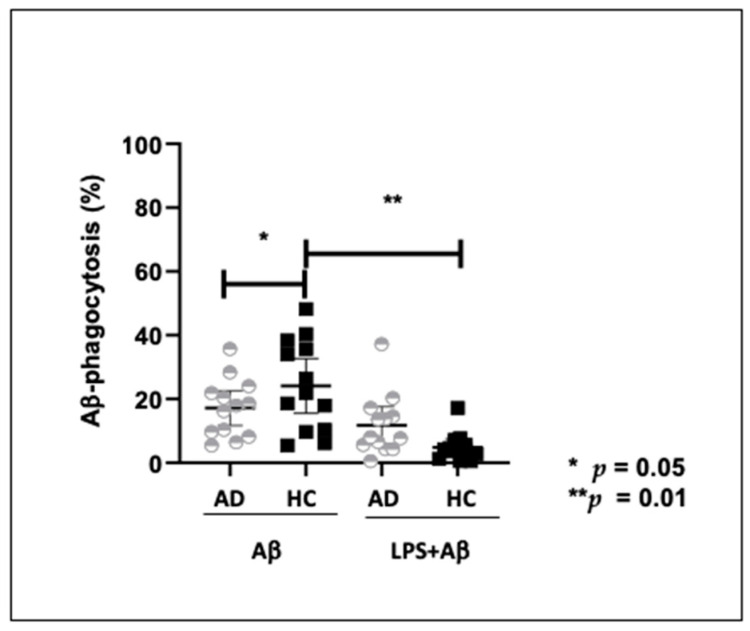
Aβ_42_-FAM-phagocytosis by Aβ- or Aβ+ LPS- stimulated monocyte of 13 Alzheimer’s Disease patients (AD) and 12 healthy controls (HC). Results are expressed as the percentage of monocytes phagocyting Aβ42-FAM. Analysis was performed by FlowSight. Mann Whitney test was used to compare the phagocytosis percentage in AD and HC and Wilcoxon matched-pair test for comparison before and after LPS stimulus Data are summarized as median and Interquartile (IQR) (25° and 75° percentile). *p* values of less than 0.05 were considered significant (* *p* = 0.05; ** *p* = 0.01).

**Figure 5 ijms-24-08626-f005:**
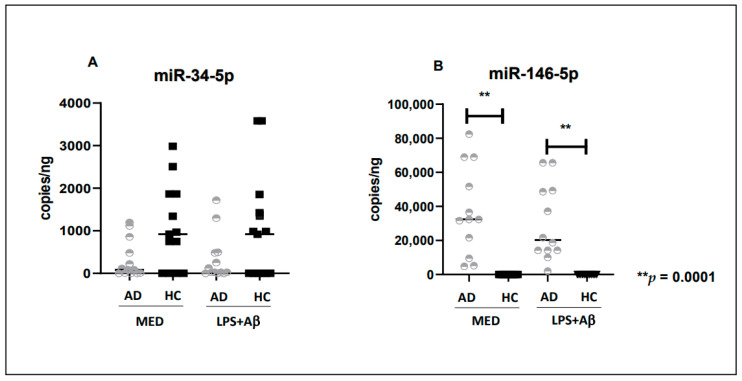
miR-34a-5p (**A**) and miR-146a-5p (**B**) expression in unstimulated (MED) or LPS + Aβ_42_ stimulated-PBMC from 12 Alzheimer’s Disease patients (AD) and 12 healthy controls (HC). The Mann Whitney test was used to compare the mRNA or miRNA expression value in AD and HC and the Wilcoxon matched-pair test for comparison before and after stimuli. For ddPCR analysis, the QuantaSoft software version 1.7.4.0917 (Bio-Rad, Hercules, CA, USA) was used to quantify mRNA and miRNAs copies. Data are summarized as median and Interquartile (IQR) (25° and 75° percentile). *p* values of less than 0.05 were considered significant ** *p* = 0.0001.

**Table 1 ijms-24-08626-t001:** Demographic and clinical data of Alzheimer’s disease patients (AD) and Healthy Control (HC). ^a^ = means, ^b^ = standard-deviation (SD).

	AD	HC
N	15	12
Gender (M:F)	09:12	04:06
Age (years)	77 ^a^ ± 5.9 ^b^	71 ^a^ ± 6.2 ^b^
Level of education (years)	8.25 ^a^ ± 2.71 ^b^	7.62 ^a^ ± 3.62 ^b^
MMSE (Baseline)	19 ^a^ ± 2.9 ^b^	>28
ApoE 4 cariers (%)	20 ^a^	20 ^a^
Ab (pg/mL)	517.75 ^a^ ± 132.71 ^b^	-
P-tau (pg/mL)	661.96 ^a^ ± 367.45 ^b^	-
Total-tau (pg/mL)	86.25 ^a^ ± 31.08 ^b^	-

## Data Availability

The raw data supporting the conclusions of this article will be made available by the authors, without undue reservation, to any qualified researcher.

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
