# Peer review of "TREM2 Expression and Amyloid-Beta Phagocytosis in Alzheimer’s Disease"

_ijms, 2023, doi:10.3390/ijms24108626_

Round 1

Reviewer 1 Report

The manuscript by La Rosa et al is an interesting study that describes the levels of the peripheral TREM2 expression and the relationship with Alzheimer´s disease. Based on the results authors highlight the relevance of the peripheral responses in neurodegeneration, which are crucial in the complete onset and progression of such diseases. For this, human samples from patients and healthy people have been used, which improves the quality of the results. In addition, some in vitro experiments with such samples have been adequately performed. However, altogether seem to be a confirmatory study as the correlation between higher sTREM2 and Alzheimer´s scores and between reduced TREM2 and diminished AB42 phagocytosis has been previously demonstrated. Therefore, the novelty of this study should be more clarified and underlined. Besides this, some points should be addressed to improve the quality of the manuscript:

1.- English should be edited paying attention to typographical mistakes.

2.- Table 1 should be showed before Figure 1, as it is described in the main text.

3.- The number of samples in each figure seems different. Please, include it in the figure legend. If that number is different, please explain the criteria to eliminate or include samples. In the in vitro experiments of Figure 4, what represent each symbol? a sample? Has this experiment performed in duplicate?...please, explain.

4.- Correlation between figure 2, 3 and 4 could better showed if they would include as different panels in the same figure.

5.- X axis in Fig. 2 and Fig. 5 is different than in Fig. 3. This is confusing. Please, homogenize.

6.- Please, include more information for the description of the Y axis for figure 4 (Phagocytosis, %; phagocytic cells, %; % of cells with AB?).

7.- The results show that higher % of CD14+TREM2+ in unstimulated HC correlates with higher phagocytosis compared to unstimulated AD. However, how the authors can explain the situation for LPS+AB stimulated AD?...In this case, although CD14+TREM2+ % is higher in stimulated AD samples, the % of phagocytosis is lower than for unstimulated AD. Please, explain. Even, % of TREM2 positive cells in unstimulated AD is similar than for stimulated HC but AD samples show higher phagocytosis. Why?...

8.- Figure 5: although levels of TREM2 are significantly different between unstimulated and stimulated HC, miR-34 is similar...Then, please explained better what the relevance of miR-34 in TREM2 expression regulation could be.

English should be edited paying attention to typographical mistakes.

Author Response

The manuscript by La Rosa et al is an interesting study that describes the levels of the peripheral TREM2 expression and the relationship with Alzheimer´s disease. Based on the results authors highlight the relevance of the peripheral responses in neurodegeneration, which are crucial in the complete onset and progression of such diseases. For this, human samples from patients and healthy people have been used, which improves the quality of the results. In addition, some in vitro experiments with such samples have been adequately performed. However, altogether seem to be a confirmatory study as the correlation between higher sTREM2 and Alzheimer´s scores and between reduced TREM2 and diminished AB42 phagocytosis has been previously demonstrated. Therefore, the novelty of this study should be more clarified and underlined. Besides this, some points should be addressed to improve the quality of the manuscript:

Q1. English should be edited paying attention to typographical mistakes.

A1:Thanks for the suggestions,  the text will be submitted to an English editing service.

Q2. Table 1 should be showed before Figure 1, as it is described in the main text.

A2:Sorry for oversight, it is a formatting system of the submission site. In the revised version of the manuscript the problem has been solved.

Q3.The number of samples in each figure seems different. Please, include it in the figure legend. If that number is different, please explain the criteria to eliminate or include samples. In the in vitro experiments of Figure 4, what represent each symbol? a sample? Has this experiment performed in duplicate?...please, explain.

A3:Thank you for raising this point, the number of samples included in the study and whose epidemiological and clinical data we collected and represented in table 1 were 15 patients AD and 12 controls. However, in the different methods included in the study, such as cell culture, RNA extraction or cell marking, some samples were eliminated due to low quality of the material. Therefore, following your suggestion, in the revised manuscript, the number of samples represented in the figure was included in the figure legend.

As regard the Figure 4, both the figure and the legend were modified in a revised text.

Q4. Correlation between figure 2, 3 and 4 could better showed if they would include as different panels in the same figure.

A4. Thank you for the suggestion, but the pictures would be too small. Therefore I have added additional material, especially supplementary figureS1 and S2 which I have included in the same panel fig 2,3 and 4,5. 

Q5.- X axis in Fig. 2 and Fig. 5 is different than in Fig. 3. This is confusing. Please, homogenize.

A5: Sorry for the mistakes, all figures have been homogenize in the revised version.

Q6.- Please, include more information for the description of the Y axis for figure 4 (Phagocytosis, %; phagocytic cells, %; % of cells with AB?).

A6: Thanks, the Figure 4 has been changed.

Q7: The results show that higher % of CD14+TREM2+ in unstimulated HC correlates with higher phagocytosis compared to unstimulated AD. However, how the authors can explain the situation for LPS+AB stimulated AD?...In this case, although CD14+TREM2+ % is higher in stimulated AD samples, the % of phagocytosis is lower than for unstimulated AD. Please, explain. Even, % of TREM2 positive cells in unstimulated AD is similar than for stimulated HC but AD samples show higher phagocytosis. Why?...

A7:Thanks for your comment. Our interpretation of this result, also based on literature data, was discussed as follows:

Line 312 “A-phagocytosis was down-regulated in LPS +A stimulated monocytes of HC, indicating that in vitro inflammatory conditions down-regulate such process. Previous results indicated that the reduction of TREM2 expression in microglia and macrophages results in a decreased phagocytosis of apoptotic neurons [77-80], cellular debris [77] and bacteria or bacterial products [81-83]; notably, increasing TREM2 expression was associated with a more efficient phagocytosis of these substrates [77,81,74-86]. TREM2 mRNA expression is modulated by inflammation. Thus, in vitro results showed that an anti-inflammatory milieu results in an upregulation of TREM2 expression [87], while pro-inflammatory proteins, such as TNFα, IL1β or lipopolysaccharide (LPS) decrease TREM2 expression [88-90]. Opposite results were described in vivo, as an increased expression of TREM2 was observed both in microglial cells and in hyppocampus of post-mortem brain of AD patients [91-94] and in mouse models of amyloid and tau pathology [85, 95-98], possibly being associated with the recruitment of microglia into amyloid plaques [94,100]. Acute inflammatory conditions, as those mimicked by in vitro studies, could thus reduce TREM2 expression, while the chronic inflammation observed in pathological conditions, such as AD, would results in increased TREM2 expression, possibly in the attempt to augment phagocytosis.”

Q8: Figure 5: although levels of TREM2 are significantly different between unstimulated and stimulated HC, miR-34 is similar...Then, please explained better what the relevance of miR-34 in TREM2 expression regulation could be.

A8: Thanks for focus this point. Epigenetic mechanisms have been previously described to regulate TREM2 gene expression and/or to be altered in the AD context. Non-coding RNAs (miRna) also participate in the epigenetic modulation of gene expression. In particular, miRNA-34a has been reported to target TREM2 mRNA 3’-untranslated region (UTR) in human AD hippocampal tissue and in primary microglia cell cultures, so it appears that miRNA-34a is involved in the down-regulation of TREM2 [1-3]. Nevertheless, Trem2 is sensitive to pro-inflammatory signaling, at least in an in vitro context [4,5]. Was reported that anti-inflammatory cytokines such as interleukin-4 (IL-4) and IL-13 [6] can upregulate Trem2 expression. Whereas Trem2 is downregulated by NF-κB-mediated miRNA-34a [7], lipopolysaccharides (LPS)-induced pro-inflammatory signaling [8], as well as pro-inflammatory cytokines [9]. In addition, the TREM2 gene may be a source of epigenetic regulators aimed at self-regulation or modulation of other genes’ expression. Recently, circRNAs have emerged as interesting molecules that deserve to be investigated as epigenetic regulators[10]. circRNAs are a particular type of non-coding RNAs that are generated by a process known as back-splicing from a pre-mRNA. As a consequence of back-splicing, circRNAs form covalently close loops, which make them resistant to the action of RNase and are, therefore, very stable molecules.

To better specify this point in the text, we modified the discussion as follows:

Line 289:   So far, the leading factors that induce TREM2 upregulation in AD-related conditions remain unclear. One possibility is that the components of Aβ plaques synergistically regulate TREM2. This is supported by a study from Neher’s group that TREM2 upregulation is triggered in microglia during their migration to plaques [66]. Another possibility is that TREM2 was induced by multiple factors of neuroinflammation [67]. Although levels of TREM2 are significantly different between unstimulated and stimulated HC, miR-34 is similar. Was reported that whereas TREM2 is downregulated by NF-κB-mediated miRNA-34a , lipopolysaccharides (LPS)-induced pro-inflammatory signaling , as well as pro-inflammatory cytokines . In addition, the TREM2 gene may be a source of epigenetic regulators aimed at self-regulation or modulation of other genes’ expression [69-73]. Recently, circRNAs have emerged as interesting molecules that deserve to be investigated as epigenetic regulators [74].”

Reference for Question 8

  1. Mehler MF, Mattick JS. Noncoding RNAs and RNA editing in brain development, functional diversification and neurological disease. Physiol Rev. 2007; 87:799–823.
  2. Smith A.R., Smith R.G., Condliffe D., Hannon E., Schalkwyk L., Mill J., Lunnon K. Increased DNA methylation near TREM2 is consistently seen in the superior temporal gyrus in Alzheimer’s disease brain. Neurobiol. Aging. 2016;47:35–40.
  3. Celarain N., Sanchez-Ruiz de Gordoa J., Zelaya M.V., Roldan M., Larumbe R., Pulido L., Echavarri C., Mendioroz M. TREM2 upregulation correlates with 5-hydroxymethycytosine enrichment in Alzheimer’s disease hippocampus.  Epigenetics. 2016;8:37.
  4. Zhao Y., Bhattacharjee S., Jones B. M., Dua P., Alexandrov P. N., Hill J. M., et al. (2013). Regulation of TREM2 expression by an NF-êB-sensitive miRNA-34a. Neuroreport 24, 318–323 10.1097/WNR.0b013e32835fb6b0 
  5. Cardoso AL, Guedes JR, de Lima MC. Role of microRNAs in the regulation of innate immune cells under neuroinflammatory conditions. Curr Opin Pharmacol. 2016; 26:1–9.
  6. . Zhai Q., Li F., Chen X., Jia J., Sun S., Zhou D., Ma L., Jiang T., Bai F., Xiong L., et al. Triggering Receptor Expressed on Myeloid Cells 2, a Novel Regulator of Immunocyte Phenotypes, Confers Neuroprotection by Relieving Neuroinflammation. 2017;127:98–110. doi: 10.1097/ALN.0000000000001628. 
  7. Zhao Y., Bhattacharjee S., Jones B.M., Dua P., Alexandrov P.N., Hill J.M., Lukiw W.J. Regulation of TREM2 expression by an NF-small ka, CyrillicB-sensitive miRNA-34a. Neuroreport. 2013;24:318–323. doi: 10.1097/WNR.0b013e32835fb6b0
  8. Ye P., Xu D., Xu J., Liu G., Huang S., Zhang W., Zheng P., Li J., Huang J. TREM-2 negatively regulates LPS-mediated inflammatory response in rat bone marrow-derived MSCs. Mol. Med. Rep. 2017;16:4777–4783. doi: 10.3892/mmr.2017.7212. 
  9. Lee B., Kim T.H., Jun J.B., Yoo D.H., Woo J.H., Choi S.J., Lee Y.H., Song G.G., Sohn J., Park-Min K.H., et al. Direct inhibition of human RANK+ osteoclast precursors identifies a homeostatic function of IL-1beta.  Immunol. 2010;185:5926–5934. doi: 10.4049/jimmunol.1001591
  10. Urdánoz-Casado, A.; de Gordoa, J.S.-R.; Robles, M.; Roldan, M.; Zelaya, M.V.; Blanco-Luquin, I.; Mendioroz, M. Profile of TREM2-Derived circRNA and mRNA Variants in the Entorhinal Cortex of Alzheimer’s Disease Patients. J. Mol.Sci.2022,23,7682. https:// doi.org/10.3390/ijms23147682

The manuscript by La Rosa et al is an interesting study that describes the levels of the peripheral TREM2 expression and the relationship with Alzheimer´s disease. Based on the results authors highlight the relevance of the peripheral responses in neurodegeneration, which are crucial in the complete onset and progression of such diseases. For this, human samples from patients and healthy people have been used, which improves the quality of the results. In addition, some in vitro experiments with such samples have been adequately performed. However, altogether seem to be a confirmatory study as the correlation between higher sTREM2 and Alzheimer´s scores and between reduced TREM2 and diminished AB42 phagocytosis has been previously demonstrated. Therefore, the novelty of this study should be more clarified and underlined. Besides this, some points should be addressed to improve the quality of the manuscript:

Q1. English should be edited paying attention to typographical mistakes.

A1:Thanks for the suggestions,  the text will be submitted to an English editing service.

Q2. Table 1 should be showed before Figure 1, as it is described in the main text.

A2:Sorry for oversight, it is a formatting system of the submission site. In the revised version of the manuscript the problem has been solved.

Q3.The number of samples in each figure seems different. Please, include it in the figure legend. If that number is different, please explain the criteria to eliminate or include samples. In the in vitro experiments of Figure 4, what represent each symbol? a sample? Has this experiment performed in duplicate?...please, explain.

A3:Thank you for raising this point, the number of samples included in the study and whose epidemiological and clinical data we collected and represented in table 1 were 15 patients AD and 12 controls. However, in the different methods included in the study, such as cell culture, RNA extraction or cell marking, some samples were eliminated due to low quality of the material. Therefore, following your suggestion, in the revised manuscript, the number of samples represented in the figure was included in the figure legend.

As regard the Figure 4, both the figure and the legend were modified in a revised text.

Q4. Correlation between figure 2, 3 and 4 could better showed if they would include as different panels in the same figure.

A4. Thank you for the suggestion, but the pictures would be too small. Therefore I have added additional material, especially supplementary figureS1 and S2 which I have included in the same panel fig 2,3 and 4,5. 

Q5.- X axis in Fig. 2 and Fig. 5 is different than in Fig. 3. This is confusing. Please, homogenize.

A5: Sorry for the mistakes, all figures have been homogenize in the revised version.

Q6.- Please, include more information for the description of the Y axis for figure 4 (Phagocytosis, %; phagocytic cells, %; % of cells with AB?).

A6: Thanks, the Figure 4 has been changed.

Q7: The results show that higher % of CD14+TREM2+ in unstimulated HC correlates with higher phagocytosis compared to unstimulated AD. However, how the authors can explain the situation for LPS+AB stimulated AD?...In this case, although CD14+TREM2+ % is higher in stimulated AD samples, the % of phagocytosis is lower than for unstimulated AD. Please, explain. Even, % of TREM2 positive cells in unstimulated AD is similar than for stimulated HC but AD samples show higher phagocytosis. Why?...

A7:Thanks for your comment. Our interpretation of this result, also based on literature data, was discussed as follows:

Line 312 “A-phagocytosis was down-regulated in LPS +A stimulated monocytes of HC, indicating that in vitro inflammatory conditions down-regulate such process. Previous results indicated that the reduction of TREM2 expression in microglia and macrophages results in a decreased phagocytosis of apoptotic neurons [77-80], cellular debris [77] and bacteria or bacterial products [81-83]; notably, increasing TREM2 expression was associated with a more efficient phagocytosis of these substrates [77,81,74-86]. TREM2 mRNA expression is modulated by inflammation. Thus, in vitro results showed that an anti-inflammatory milieu results in an upregulation of TREM2 expression [87], while pro-inflammatory proteins, such as TNFα, IL1β or lipopolysaccharide (LPS) decrease TREM2 expression [88-90]. Opposite results were described in vivo, as an increased expression of TREM2 was observed both in microglial cells and in hyppocampus of post-mortem brain of AD patients [91-94] and in mouse models of amyloid and tau pathology [85, 95-98], possibly being associated with the recruitment of microglia into amyloid plaques [94,100]. Acute inflammatory conditions, as those mimicked by in vitro studies, could thus reduce TREM2 expression, while the chronic inflammation observed in pathological conditions, such as AD, would results in increased TREM2 expression, possibly in the attempt to augment phagocytosis.”

Q8: Figure 5: although levels of TREM2 are significantly different between unstimulated and stimulated HC, miR-34 is similar...Then, please explained better what the relevance of miR-34 in TREM2 expression regulation could be.

A8: Thanks for focus this point. Epigenetic mechanisms have been previously described to regulate TREM2 gene expression and/or to be altered in the AD context. Non-coding RNAs (miRna) also participate in the epigenetic modulation of gene expression. In particular, miRNA-34a has been reported to target TREM2 mRNA 3’-untranslated region (UTR) in human AD hippocampal tissue and in primary microglia cell cultures, so it appears that miRNA-34a is involved in the down-regulation of TREM2 [1-3]. Nevertheless, Trem2 is sensitive to pro-inflammatory signaling, at least in an in vitro context [4,5]. Was reported that anti-inflammatory cytokines such as interleukin-4 (IL-4) and IL-13 [6] can upregulate Trem2 expression. Whereas Trem2 is downregulated by NF-κB-mediated miRNA-34a [7], lipopolysaccharides (LPS)-induced pro-inflammatory signaling [8], as well as pro-inflammatory cytokines [9]. In addition, the TREM2 gene may be a source of epigenetic regulators aimed at self-regulation or modulation of other genes’ expression. Recently, circRNAs have emerged as interesting molecules that deserve to be investigated as epigenetic regulators[10]. circRNAs are a particular type of non-coding RNAs that are generated by a process known as back-splicing from a pre-mRNA. As a consequence of back-splicing, circRNAs form covalently close loops, which make them resistant to the action of RNase and are, therefore, very stable molecules.

To better specify this point in the text, we modified the discussion as follows:

Line 289:   So far, the leading factors that induce TREM2 upregulation in AD-related conditions remain unclear. One possibility is that the components of Aβ plaques synergistically regulate TREM2. This is supported by a study from Neher’s group that TREM2 upregulation is triggered in microglia during their migration to plaques [66]. Another possibility is that TREM2 was induced by multiple factors of neuroinflammation [67]. Although levels of TREM2 are significantly different between unstimulated and stimulated HC, miR-34 is similar. Was reported that whereas TREM2 is downregulated by NF-κB-mediated miRNA-34a , lipopolysaccharides (LPS)-induced pro-inflammatory signaling , as well as pro-inflammatory cytokines . In addition, the TREM2 gene may be a source of epigenetic regulators aimed at self-regulation or modulation of other genes’ expression [69-73]. Recently, circRNAs have emerged as interesting molecules that deserve to be investigated as epigenetic regulators [74].”

Reference for Question 8

  1. Mehler MF, Mattick JS. Noncoding RNAs and RNA editing in brain development, functional diversification and neurological disease. Physiol Rev. 2007; 87:799–823.
  2. Smith A.R., Smith R.G., Condliffe D., Hannon E., Schalkwyk L., Mill J., Lunnon K. Increased DNA methylation near TREM2 is consistently seen in the superior temporal gyrus in Alzheimer’s disease brain. Neurobiol. Aging. 2016;47:35–40.
  3. Celarain N., Sanchez-Ruiz de Gordoa J., Zelaya M.V., Roldan M., Larumbe R., Pulido L., Echavarri C., Mendioroz M. TREM2 upregulation correlates with 5-hydroxymethycytosine enrichment in Alzheimer’s disease hippocampus.  Epigenetics. 2016;8:37.
  4. Zhao Y., Bhattacharjee S., Jones B. M., Dua P., Alexandrov P. N., Hill J. M., et al. (2013). Regulation of TREM2 expression by an NF-êB-sensitive miRNA-34a. Neuroreport 24, 318–323 10.1097/WNR.0b013e32835fb6b0 
  5. Cardoso AL, Guedes JR, de Lima MC. Role of microRNAs in the regulation of innate immune cells under neuroinflammatory conditions. Curr Opin Pharmacol. 2016; 26:1–9.
  6. . Zhai Q., Li F., Chen X., Jia J., Sun S., Zhou D., Ma L., Jiang T., Bai F., Xiong L., et al. Triggering Receptor Expressed on Myeloid Cells 2, a Novel Regulator of Immunocyte Phenotypes, Confers Neuroprotection by Relieving Neuroinflammation. 2017;127:98–110. doi: 10.1097/ALN.0000000000001628. 
  7. Zhao Y., Bhattacharjee S., Jones B.M., Dua P., Alexandrov P.N., Hill J.M., Lukiw W.J. Regulation of TREM2 expression by an NF-small ka, CyrillicB-sensitive miRNA-34a. Neuroreport. 2013;24:318–323. doi: 10.1097/WNR.0b013e32835fb6b0
  8. Ye P., Xu D., Xu J., Liu G., Huang S., Zhang W., Zheng P., Li J., Huang J. TREM-2 negatively regulates LPS-mediated inflammatory response in rat bone marrow-derived MSCs. Mol. Med. Rep. 2017;16:4777–4783. doi: 10.3892/mmr.2017.7212. 
  9. Lee B., Kim T.H., Jun J.B., Yoo D.H., Woo J.H., Choi S.J., Lee Y.H., Song G.G., Sohn J., Park-Min K.H., et al. Direct inhibition of human RANK+ osteoclast precursors identifies a homeostatic function of IL-1beta.  Immunol. 2010;185:5926–5934. doi: 10.4049/jimmunol.1001591
  10. Urdánoz-Casado, A.; de Gordoa, J.S.-R.; Robles, M.; Roldan, M.; Zelaya, M.V.; Blanco-Luquin, I.; Mendioroz, M. Profile of TREM2-Derived circRNA and mRNA Variants in the Entorhinal Cortex of Alzheimer’s Disease Patients. J. Mol.Sci.2022,23,7682. https:// doi.org/10.3390/ijms23147682

Reviewer 2 Report

1.       Abstract: Provide some background information: It would be helpful to include some information on Alzheimer's disease (AD) and the importance of studying immune receptors like TREM2 in the context of the disease. Clarify methods and results: The methods used to collect and analyze the data should be clearly described, including sample size, statistical analysis, and how Aβ42-phagocytosis was measured. Results should be clearly presented with relevant statistics and p-values.

Discuss implications: The abstract mentions that the results support a role for TREM2-expressing cells in AD pathogenesis and suggest that TREM2 analysis could be a diagnostic and prognostic biomarker for AD. These implications should be further discussed and expanded upon.

Highlight limitations: The abstract should also mention any limitations of the study, such as the small sample size, and the need for further research to confirm these findings

2.       The introduction is quite lengthy and contains a lot of technical jargon and abbreviations that may be difficult for some readers to understand. The sentences are also quite long and could benefit from being broken down into smaller, more concise statements. Additionally, the introduction jumps straight into discussing the TREM2 protein without providing much context or background information on the topic. A more effective introduction could provide a brief overview of the relevance of neuroinflammation and neurodegeneration, followed by a more general introduction to the innate immune system and the role of microglia in the brain, before delving into the specific role of TREM2. Finally, the introduction could benefit from a clearer statement of the research question or hypothesis being investigated.

3.       Section 2.4, Small sample size: The study does not mention the number of AD patients and controls included in the analysis, but it is important to ensure that the sample size is large enough to make statistically significant conclusions.

Lack of blinding: The study does not mention if the individuals performing the flow cytometry analysis were blinded to the identity of the samples, which could introduce bias in the results.

Lack of replication: The study does not mention if the results were replicated in independent experiments, which is important to establish the robustness of the findings.

Use of Aβ42: The study uses Aβ42 to stimulate immune cells, but Aβ42 is known to be neurotoxic and can induce inflammation in the brain. It is possible that the effects observed on TREM2 expression are not specific to immune activation, but rather a response to the toxic effects of Aβ42.

Lack of functional analysis: The study only examines TREM2 expression on immune cells, but it is important to also examine whether this expression translates into functional changes in immune cell activity. Without functional analysis, it is unclear what the biological significance of the observed changes in TREM2 expression is.

4.       Section, 2.6: The study is in vitro and does not account for the complex in vivo conditions, which may affect the interpretation of the results.

The study only evaluates the phagocytosis of Aβ42 in monocytes, which may not be representative of other immune cells or the entire immune response.

The study does not investigate the mechanism underlying the reduction in phagocytosis and TREM2 expression, which limits the understanding of the underlying biology.

5.       The discussion could be better structured and more focused on the main findings of the study.

The language is sometimes convoluted, which makes it difficult to understand the main points.

The authors could provide more details on the methods used to measure sTREM2 and miRNA levels and their limitations.

The discussion would benefit from more context on the current understanding of the role of TREM2 in AD and the limitations of using peripheral biomarkers for disease diagnosis and progression.

see above

Author Response

Q1: Abstract: Provide some background information: It would be helpful to include some information on Alzheimer's disease (AD) and the importance of studying immune receptors like TREM2 in the context of the disease. Clarify methods and results: The methods used to collect and analyze the data should be clearly described, including sample size, statistical analysis, and how Aβ42-phagocytosis was measured. Results should be clearly presented with relevant statistics and p-values. Discuss implications: The abstract mentions that the results support a role for TREM2-expressing cells in AD pathogenesis and suggest that TREM2 analysis could be a diagnostic and prognostic biomarker for AD. These implications should be further discussed and expanded upon. Highlight limitations: The abstract should also mention any limitations of the study, such as the small sample size, and the need for further research to confirm these findings

      The introduction is quite lengthy and contains a lot of technical jargon and abbreviations that may be difficult for some readers to understand. The sentences are also quite long and could benefit from being broken down into smaller, more concise statements. Additionally, the introduction jumps straight into discussing the TREM2 protein without providing much context or background information on the topic. A more effective introduction could provide a brief overview of the relevance of neuroinflammation and neurodegeneration, followed by a more general introduction to the innate immune system and the role of microglia in the brain, before delving into the specific role of TREM2. Finally, the introduction could benefit from a clearer statement of the research question or hypothesis being investigated.

      A1: Thanks for going thru our work with such care and thanks for your comments.  We have gone through an in-depth revision of the Manuscript (of note, we shortened the Introduction and refocused the Abstract, as per your suggestion). We hope you will be satisfied with this revised version of the manuscript.

The abstract was modified as follows: 

Abstract:Alzheimer’sDisease(AD) is the most common form of dementia; its key pathological findings include the deposition of extracellular-neurotoxic-plaques composed of amyloid-β (Aβ). AD-pathogenesis involves mechanisms that operate outside the brain, and new researches indicate that peripheral inflammation is an early event in the disease. Herein, we focus on a receptor known as triggering-receptor-expressed-on-myeloid-cells2 (TREM2), which promotes the optimal immune cells function required to attenuate AD-progression and is, therefore, a potential target as peripheral diagnostic and prognostic-biomarker for AD.

The objective of this exploratory study was to analyze:1)soluble-TREM2 (sTREM2) plasma and CSF concentration, 2)TREM2-mRNA, 3)the percentage of TREM2-expressing monocytes, and 4)the concentration of miR-146a-5p and miR-34a-5p suspected to influence TREM2 transcription. Experiments were performed on PBMC collected by 15AD patients and 12age-matched healthy-controls(HC) that were unstimulated or treated in inflammatory (LPS) conditions and Aβ42 for 24hours; Aβ42-phagocytosis was also analyzed by AMNIS FlowSight. Results although preliminary, due to limitations by the small sample-size, showed that in AD compared to HC: TREM2 expressing monocytes were reduced, plasma sTREM2 concentration and TREM2-mRNA were significantly upregulated and Aβ42-phagocytosis was diminished (for all p<0.05). miR-34a-5p expression was reduced(p = 0.02) as well in PBMC of AD , and miR-146 was only observed in AD cells (p=0.0001).”

The Introduction was modified as follows: 

“The mechanisms by which innate immune responses contribute to neuroinflammation and neurodegeneration are only partially understood. The triggering receptor expressed on myeloid cells-2 (TREM2) protein is a transmembrane protein belonging to the TREM family and is an important innate immune receptor in the brain. TREM2 is primarily expressed by microglia, myeloid cells that in the central nervous system (CNS) are involved in immune surveillance, cell-cell interactions, control of latent inflammatory reactions, and tissue debris clearance [1-5]. Thus, TREM2 plays an important role in microglial phagocytosis of apoptotic neurons,  damaged myelin, and amyloid plaques [6-13]. Furthermore, TREM2 regulates microglial biosynthetic metabolism [14], proliferation, and survival [15], cytokine release [16], and their accumulation around plaques. SNPs in the TREM2 gene modulate protein expression [17-18] and function, possibly resulting in an inactive receptor [17]. Such SNPs were identified as risk factors for Alzheimer’s disease (AD), a central nervous system disease that leads to dementia [19] and it is characterized by progressive cognitive dysfunction, memory loss, and neuroinflammation [20-27].Recently, two independent studies reported that a heterozygous rare variant in TREM2 p.R47H is significantly associated with AD [28-29] and multiple variants in the same gene increase AD risk [30-32].

According to amyloid cascade theory [33] the presence of extracellular plaques of insoluble β-amyloid peptide (Aβ) and neurofibrillary tangles (NFT) of P-tau in neuronal cytoplasm is the hallmark of AD [34]. The cerebral accumulation both extra- and intraneuronally of Aβ peptide, resulting from the imbalance between production and clearance of this protein, is the main event causing the disease [34]. In the AD brain, Aβ clearance can be mediated by drainage to perivascular spaces or by Aß uptake; it is processing by different cell types [35-36]. When activated by ligands, such as Aβ, TREM2 induces an innate immune response, which includes phagocytosis, chemotaxis, and transcriptional changes [37]. The impact of TREM2 on plaque accumulation in amyloid pathology was examined in TREM2-deficient AD murine models, but results were conflicting [6-17]. Interestingly, a later study suggested disease progression-dependent effects of TREM2 on amyloid pathology. TREM2 deficiency ameliorates amyloid pathology in the early phases of the disease, but exacerbates it late in the disease process [38]. Notably, while resident microglia plays a key role in Aβ clearance in the brain, approximately 40–60% of brain-generated Aβ is estimated to diffuse into the blood and being cleared in the periphery, implying that peripheral mechanisms play an essential role as well in Aβ clearing [37,39].

TREM2 undergoes regulated proteolytic cleavage in the H157-S158 peptide bond by two proteins, ADAM10 and ADAM17, resulting in the generation of a soluble form of the protein (sTREM2) [40-42]. sTREM2 can be measured in plasma and in cerebrospinal fluid (CSF) [43-45]. sTREM2 concentration changes dynamically during the progression of AD, peaking at the early symptomatic stages of the disease [43-46]. sTREM2 CSF concentrations was reported to correlate with neuronal injury markers, including the CSF concentration of total tau and phospho-tau [47]. Notably, TREM2 mRNA detected in peripheral monocytes and sTREM2 serum concentration were shown to directly correlate with clinical parameters for AD diagnosis[48-49]. Therefore, in this study, we investigated peripheral TREM2 expression and the concentration of 4 miRNAs (miR-146a-5p, miR-125b, miR-9-3p, miR-34a-5p), which are differentially expressed in AD and are suspected to influence the rate of TREM2 mRNA transcription [50], in AD patients and HC. Results herein show that sTREM2 plasma concentration and mRNA are increased whereas TREM2-expressing monocytes are reduced in AD. This was correlated with an impairment of Aβ42-phagocytosis. Further data indicating a direct correlation between miR-34a-5p concentration and TREM2-mRNA and sTREM2 plasma concentration suggest a possible regulatory role for this miRNA on TREM2 biosynthesis.”

Q2.  Section 2.4, Small sample size: The study does not mention the number of AD patients and controls included in the analysis, but it is important to ensure that the sample size is large enough to make statistically significant conclusions.

A2: Thanks for the note. We are aware that the small sample size is a limitation; due to the impediments of the last two years (covid restriction for frail patients), the patients recruitment  has been very difficult. Based on your suggestion, we have modified section 2.4 and the legend of figure 3 as follows:

2.4 TREM2 expressing monocytes in AD and HC

The percentage of TREM2-expressing cells was examined by flow cytometry in a subset of subjects, selected according to the availability of enough cells to run the experiment. Thus, CD14+ immune cells of 10 AD patients and 10 control were either unstimulated or stimulated with LPS and Aβ42 for 24 hours. In unstimulated conditions (medium), the percentage of TREM2 expressing monocytes was increased although not significantly in HC (median 3.2%) compared to AD patients (median 0.9 %). Interestingly, TREM-2 expression on monocytes of HC was significant reduced (p < 0.05) when cells were stimulated with LPS+Aβ42, suggesting a negative effect of inflammatory stimulation on TREM2 expression (Figure 3).”

Figure 3. TREM2-expressing CD14+ monocytes in 10 Alzheimer’s Disease patients (AD) and 10 healthy controls (HC); ……”

Q3: Lack of blinding: The study does not mention if the individuals performing the flow cytometry analysis were blinded to the identity of the samples, which could introduce bias in the results.

A3: All experiments included in the study were conducted in blind on the same sample cohort. Although, as mentioned above (Q2) for some experiments we were forced to work with a subgroup but still from the same cohort.

Q4:Lack of replication: The study does not mention if the results were replicated in independent experiments, which is important to establish the robustness of the findings.

A4: Thank you for raising this point. In our previous study (La Rosa et al, 2019; Ref 56 in the revised text) the expression of TREM2 associated with Ab-phagocytic-activity of cells , was tested in in vitro model of THP-1 monocytes cell lines. To enhance this concept, we have modified the discussion as follows:

Line 258: “Results herein confirm data obtained previously [56] and suggest that TREM2 expression is altered in AD and help to define the features of this immune receptor in peripheral blood. We observed that sTREM2 can be measured both in CSF and plasma;

Q5: Use of Aβ42: The study uses Aβ42 to stimulate immune cells, but Aβ42 is known to be neurotoxic and can induce inflammation in the brain. It is possible that the effects observed on TREM2 expression are not specific to immune activation, but rather a response to the toxic effects of Aβ42.

A5: the concentration of Ab-amyloid used to stimulate cells, was tested both through a toxicity assay (MTT) using a range of concentrations (1 to 10 mM) suggested by the company. In the revised text, was included in the Supplementary data, the Figure S4, related to the MTT assay of Ab treatment for 24h. The point 4.2 of Methods was changed as follows:

….the medium was changed and monocytes were incubated with Lypopolisacaride (LPS) (1g/ml) (Sigma-Aldrich, St. Louis, MO, US) and A42 (2.5 M) (Phoenix Pharmaceuticals, Burlingame, CA, US) (Figure S3), or cultured with Alexa Flu

Q6: Section, 2.6: The study is in vitro and does not account for the complex in vivo conditions, which may affect the interpretation of the results.

The study only evaluates the phagocytosis of Aβ42 in monocytes, which may not be representative of other immune cells or the entire immune response.

The study does not investigate the mechanism underlying the reduction in phagocytosis and TREM2 expression, which limits the understanding of the underlying biology.

A6: We are aware of the limitations involves in in vitro experimentation; certainly cell cultures can not mimic a human organism. However, it remains a fundamental step in scientific research, useful for understanding those molecular mechanisms that will be needed for in vivo experimentation. We can consider this work as a preliminary study to allow in the further studies that also includes an in vivo part. This limitation was include in  the revised text as follows:

                                             “Conclusions:

We are aware that there are limitations to this study: thus, the number of the study participants is limited, and these results need to be further investigated in large cohorts of patients. Secondly, we could not analyze these parameters in CSF of HC. Finally, in vitro experimentation: cell cultures can not mimic the human These limitations notwithstanding, our data reinforce the notion that TREM2 plays an important role in AD, shed some light on the mechanisms responsible for its regulation, and suggest that TREM2 measurement in peripheral blood could be a useful biomarker for AD diagnosis and prognosis. As future prospects, we hope to also investigate trem2 gene-related mutations on peripheral immune cells and its implications on the course of AD.”

About the phagocytosis of Aβ42, in general,  phagocytes are equipped with specialized receptors to recognize their targets, included TREM2;  this complex machinery mediates internalization and initiates an assortment of degradative mechanisms that culminate in killing and disposal of the engulfed particles. An important mechanism in AD and not fully elucidated. Most of our knowledge of the molecular mechanisms of phagocytosis has been derived from studies of phagocytosis in monocytes or macrophages (La Rosa et al. JAD. 2019), and much less is known about neutrophils, due to the fact that they are refractory to genetic manipulation by either transfection or microinjection. For this reason, our reference will be made to phagocytosis in monocytes lines; however, your observation is interesting, it proposes an excellent point to future investigation.

At first,  in this study we selected only monocytes to confirm our previous data obtained in THP-1 cells lines;  secondarily, because monocytes show microglia-like features given that the most common TREM2 researches  is conducted on mouse microglia or microglial lines . Also, monocytes, play a central role in the inflammatory response to microbial pathogens  mediated via TLRs.  In particular, LPS triggers a pro-inflammatory response, which is characterized by a production of inflammatory cytokines such as IL-1β and IL-6, via TLR2,4 (T. Kawai & S. Akira. Nat Immunol 2010). A condition that mimics an inflammatory environment that we want to recreate in an in vitro model of AD.

By means of cytofluorometer analysis, we assessed TREM2 receptor expression on the surface of monocytes; instead, Ab42- phagocytosis was assessed by AMNIS flow sight analysis. We didn’t show correlation analysis because the data were not sufficient; but the results, although taken separately, have shown in unstimulated cells of AD compared to HC: 1)a reduction in TREM2 receptor expression;2) an increase of sTREM2 in plasma ; and 3)a reduction Ab-phagocytosis. Also,4) results obtained on analyzing mean fluorescence intensity (MFI) showed that TREM2 MFI was reduced, although not significantly. These data led to the suggestion that low-level TREM2 expression may allow a reduced Ab-phagocytic capacity  by monocytes of AD patients.

To highlight this point, we added this sentence in the discussion:

Line 273: Herein we have shown that in AD sTREM2 concentration was increased concomitantly with a decrease of TREM2-expressing monocytes, a reduction of TREM2 level of receptor expression and A42-phagocytosis by such monocytes Taken together these results suggest that in vitro A42-phagocytosis is impaired in AD as a consequence of the ligation of monocytes-expressed TREM2 by the increased amounts of its soluble, decoy form

Q7: The discussion could be better structured and more focused on the main findings of the study.

The language is sometimes convoluted, which makes it difficult to understand the main points.

The authors could provide more details on the methods used to measure sTREM2 and miRNA levels and their limitations.

The discussion would benefit from more context on the current understanding of the role of TREM2 in AD

A6: Thank you for your suggestion, we have revised the discussion as follows:

“ Discussion

Alzheimer’s disease (AD) is the most common type of dementia and imposes substantial economic and social burdens [51]. Biomarkers are crucial for the accurate and early identification of AD and are a prerequisite for effective management of the disease.

Immunity and inflammation are essential processes at play throughout the whole AD process, and the related biomarkers could be part of the diagnosis.

Since the establishment of the common protocol for the AD , cerebrospinal fluid tests and positron emission tomography examinations have become widely accepted. However problems with invasiveness and high cost limit the application of the above diagnostic methods aimed at the central nervous system. Therefore, different studies suggest a focus on peripheral biomarkers in the diagnosis of AD; also the structures of biomarkers or a combinations of interacting biomarkers and the methodologies for the detection are important for the exploration of AD.

Triggering receptor expressed on myeloid cells-2 (TREM2) is a receptor in the microglial  membrane; it interacts directly with Aβ, which restricts the pathological enhancement of Aβ and tau[52]. TREM2 is expressed on cells of the myeloid lineage, including microglia and monocyte-derived macrophages [18], and plays a key role in hampering neuroinflammation by inhibiting the persistent activation of microglia, promoting phagocytosis, and clearing apoptotic neurons [53]. Because CSF concentration of sTREM2 correlates with CSF levels of total tau and phospho-tau, possibly being a biomarker for neurodegeneration [54-55], we verified whether TREM2 expression on peripheral immune cells and sTREM2 plasma concentration could be used as easily accessible peripheral biomarkers for the onset and/or progression of AD. To this end, we analyzed TREM2 RNA, its expression on monocytes as well as sTREM2 plasma concentration in a cohort of Alzheimer’s patients (AD) comparing results to those obtained in a control population of healthy subjects matched for sex and age (HC). Results herein confirm data obtained previously [56] and suggest that TREM2 expression is altered in AD and help to define the features of this immune receptor in peripheral blood. We observed that sTREM2 can be measured both in CSF and plasma; CSF concentrations being significantly higher than those seen in plasma. Thus, a higher plasmatic concentration of sTREM2 was found in our well-characterized group of AD patients compared to controls without cognitive decline. It is noteworthy that, while CSF sTREM2 levels have been extensively studied, only a few studies have investigated sTREM2 in plasma, often producing conflicting results [57-60]. Indeed, existing studies report no significant difference in the plasmatic levels of sTREM2 between AD and HC [61-62]. It has been speculated that changes in sTREM2 concentration in these biofluids are directly correlated with the entity of microglial dysfunction and neuroinflammation in AD [58]. Although it is known that the CSF sTREM2 concentration is elevated in AD [55], it remains unknown how sTREM2 impacts amyloid pathology. Nevertheless, different data suggest that sTREM2 possesses important biological and pathological properties other than acting as a decoy receptor that opposes full-length TREM2 signaling .

Herein we have shown that in AD sTREM2 concentration was increased concomitantly with a decrease of TREM2-expressing monocytes, a reduction of TREM2 level of receptor expression and A42-phagocytosis by such monocytes Taken together these results suggest that in vitro A42-phagocytosis is impaired in AD as a consequence of the ligation of monocytes-expressed TREM2 by the increased amounts of its soluble, decoy form.

We also evaluated TREM2 mRNA expression and its possible miRNA targets, including miRNA-34a-5p. The miRNA-34a-mediated down-regulation of TREM2 appears to be recognition feature within the 299 nt TREM2 mRNA 3′-UTR region. Indeed, it is known that epigenetic mechanisms involving miRNA-34a up-regulation and consequent down-regulation of TREM2 expression may drive the progressive extinction of the phagocytic response that in turn contributes to dysfunctional innate-immunity, amyloidogenesis and inflammatory neurodegeneration [63]. Increased TREM2 mRNA expression was described in peripheral blood mononuclear cells from mild cognitive-impaired patients that later converted to AD [64], as well as in blood from subjects with an increased risk to develop dementia [65]. Here, we found a higher TREM 2-mRNA expression in monocytes of AD compared to HC; in the same individuals miRNA-34a-5p expression was reduced. So far, the leading factors that induce TREM2 upregulation in AD-related conditions remain unclear. One possibility is that the components of Aβ plaques synergistically regulate TREM2. This is supported by a study from Neher’s group that TREM2 upregulation is triggered in microglia during their migration to plaques [66]. Another possibility is that TREM2 was induced by multiple factors of neuroinflammation [67]. Although levels of TREM2 are significantly different between unstimulated and stimulated HC, miR-34 is similar. Was reported that whereas TREM2 is downregulated by NF-κB-mediated miRNA-34a , lipopolysaccharides (LPS)-induced pro-inflammatory signaling , as well as pro-inflammatory cytokines . In addition, the TREM2 gene may be a source of epigenetic regulators aimed at self-regulation or modulation of other genes’ expression [69-73]. Recently, circRNAs have emerged as interesting molecules that deserve to be investigated as epigenetic regulators [74].

Besides miRNA-34a-5p, the only other mRNA that could be measured in our study was miR-146a-5p, a key regulator of the immune response [75] that has been implicated in multiple neuroinflammatory processes, including AD [76]. Thus, studies in murine models of AD and ex vivo results showed that miR-146a expression increases with disease progression and correlates with senile plaque density and synaptic pathology [75-76]. Our results showed that miR-146a-5p could be detected in AD patients alone. This mRNA was shown to reduce TREM2 expression and A-phagocytosis; all these findings were present in our study. These results support the idea that increased miR-146a levels could downregulate the expression of TREM2, leading to reduced Aβ clearance [76]. Moreover, our study results confirmed the idea that increased miR146 expression correlates with reduction of A-phagocytosis in AD patients compared to HC.

A-phagocytosis was down-regulated in LPS +A stimulated monocytes of HC, indicating that in vitro inflammatory conditions down-regulate such process. Previous results indicated that the reduction of TREM2 expression in microglia and macrophages results in a decreased phagocytosis of apoptotic neurons [77-80], cellular debris [77] and bacteria or bacterial products [81-83]; notably, increasing TREM2 expression was associated with a more efficient phagocytosis of these substrates [77,81,74-86]. TREM2 mRNA expression is modulated by inflammation. Thus, in vitro results showed that an anti-inflammatory milieu results in an upregulation of TREM2 expression [87], while pro-inflammatory proteins, such as TNFα, IL1β or lipopolysaccharide (LPS) decrease TREM2 expression [88-90]. Opposite results were described in vivo, as an increased expression of TREM2 was observed both in microglial cells and in hyppocampus of post-mortem brain of AD patients [91-94] and in mouse models of amyloid and tau pathology [85, 95-98], possibly being associated with the recruitment of microglia into amyloid plaques [94,100]. Acute inflammatory conditions, as those mimicked by in vitro studies, could thus reduce TREM2 expression, while the chronic inflammation observed in pathological conditions, such as AD, would results in increased TREM2 expression, possibly in the attempt to augment phagocytosis.

To summarize results presented herein, in AD patients compared to the HC: 1) the expression of TREM2 is increased, that of its negative feedback inhibitor mir34a-5p is reduced;  2)the one of miR-146a, a TREM2 down-regulator, is increased; 3) TREM2 expression on monocyte is reduced whereas sTREM2 plasma concentration is increased, and 4) A-phagocytosis is decreased. Taken together these results support an important role for miRNA-34a-5p and miR-146a in regulating TREM2 expression. These data also allow the speculation that in AD a significant amount of the TREM2 that is translated into protein undergoes cleavage into sTREM2; it is released into biological fluids, not being available as a cellular receptor to facilitate Aphagocytosis.

Conclusions:

We are aware that there are limitations to this study: thus, the number of the study participants is limited, and these results need to be further investigated in large cohorts of patients. Secondly, we could not analyze these parameters in CSF of HC. Finally, in vitro experimentation: cell cultures cannot mimic the human. These limitations notwithstanding, our data reinforce the notion that TREM2 plays an important role in AD, shed some light on the mechanisms responsible for its regulation, and suggest that TREM2 measurement in peripheral blood could be a useful biomarker for AD diagnosis and prognosis. As future prospects, we hope to also investigate trem2 gene-related mutations on peripheral immune cells and its implications on the course of AD.”

Reviewer 3 Report

My suggestions:

1. In the introduction, I would mention a few examples of mutations, discovered in the TREM2 gene, which may impact AD risk.

2. In Table 1,  were the APOE E4 allele carriers only the homozygous E4 allele? Or did they carry E3/E4 or E2/E4 was also included?

3. Did the patients carry any mutation in TREM2? 

 4. I would suggest including the figures and tables in the paragraph when they were mentioned first. 

5. Was there any correlation found between TREM2 expression, and plasma amyloid beta levels or Ab42/40 rato? How about the CSF TREM2 expression and CSF-Tau? Was there any correlation between them? 

6. Were there any other miRNAs identified before, and which expression may be changed in the case of TREM2 expressing monocytes?

7. I would add a figure in the discussion, which shows through which possible pathways miRNA-34a-5p and miR-146a-5p could regulate in case of altered TREM2 expression. 

English seems fine. 

Author Response

REVIEWER 3

Q1: In the introduction, I would mention a few examples of mutations, discovered in the TREM2 gene, which may impact AD risk.

A1: Thanks for your comment, as you have suggested, the introduction has been modified as follows:

Line 46 “Such SNPs were identified as risk factors for Alzheimer’s disease (AD), a central nervous system disease that leads to dementia [19] and it is characterized by progressive cognitive dysfunction, memory loss, and neuroinflammation [20-27].Recently, two independent studies reported that a heterozygous rare variant in TREM2 p.R47H is significantly associated with AD [28-29] and multiple variants in the same gene increase AD risk [30-32].”

Q2: In Table 1,  were the APOE E4 allele carriers only the homozygous E4 allele? Or did they carry E3/E4 or E2/E4 was also included?

A2: This is a good point, in this study we evaluated all three common alleles, ε2, ε3, and ε4 of APOE gene, but  in the table we only reported the frequency of apoE4 allele. As your suggestion we have include the details of the analysis in the text as follows: 

“2.1 Patients and control

The demographic and clinical characteristics of the individuals enrolled in the study are summarized in Table 1. No differences were observed in gender, age, or years of education or genotype distribution of ApoE 4 carrier. The allele frequency of apoE 4 that was found in AD patients and in the control population (HC) was 20%; in detail: 4 of the 15 patients carried the Apo E4 allele (Apo E4+): 2 were E4+/E4+ homozygote and 2 were E4+/E3+ heterozygote. Among Apo E4−/E4− subjects, all 12 patients carrier the E3+/E3+ genotype. In HC:  3 of 12 subjects included in the study were E4+/E3+ heterozygote and 1 was E2+/E4+ ; 8 were E3+/E3+……..”

Q3: Did the patients carry any mutation in TREM2?

A3: Thanks for your encouraging comments,  in the current study we did not evaluate any mutation in TREM2. Because the sample size would not have allowed for a genetic survey, it is certainly a good point that we could explore further in a larger cohort of future studies. Based on your suggestion, we have included a note for the future perspectives as follows:

                                              “Conclusions:

We are aware that there are limitations to this study: thus, the number of the study participants is limited, and these results need to be further investigated in large cohorts of patients. Secondly, we could not analyze these parameters in CSF of HC. Finally, in vitro experimentation: cell cultures cannot mimic the human. These limitations notwithstanding, our data reinforce the notion that TREM2 plays an important role in AD, shed some light on the mechanisms responsible for its regulation, and suggest that TREM2 measurement in peripheral blood could be a useful biomarker for AD diagnosis and prognosis. As future prospects, we hope to also investigate trem2 gene-related mutations on peripheral immune cells and its implications on the course of AD.

Q4: I would suggest including the figures and tables in the paragraph when they were mentioned first. 

A4: Thank you for your suggestion, we have carefully reviewed the text.

Q5: Was there any correlation found between TREM2 expression, and plasma amyloid beta levels or Ab42/40 rato? How about the CSF TREM2 expression and CSF-Tau? Was there any correlation between them? 

A5: Good point, we tried to correlate the data but the results were not significant, probably due to the sample size.

Q6: Were there any other miRNAs identified before, and which expression may be changed in the case of TREM2 expressing monocytes?

A6: Thanks for your comment, in this work, we investigated peripheral TREM2 expression and the concentration of 4 miRNAs (miR-146a-5p, miR-125b, miR-9-3p, miR-34a-5p), which are differentially expressed in AD and are suspected to influence the rate of TREM2 mRNA transcription [Pogue, A.I. et al 2018, reference number….in the text], in AD patients and HC. Of the four miRNAs mentioned, only miR-34-5p and miR-146-5p were found to be significant, so our discussion focused only on these two miRnas.

Notably, most of the TREM2  studies have been conducted in vivo on mouse, microglia cell or hippocampal or cortex tissue. Also, the expression of mRNA-TREM2 in monocytes of AD patients has not yet been  investigated. Even more, no association was reported with the deregulation of specific immune-related miRNAs  differentially expressed in blood-derived monocyte and the expression and processing of TREM2. According to a previus study (Nan Hu et al _doi: 10.3233/JAD-130854), which evaluated TREM2 mRNA and protein expressions in peripheral blood from AD patients and healthy controls, data indicate that TREM2 might serve as a novel noninvasive biomarker for AD diagnosis.

Q7. I would add a figure in the discussion, which shows through which possible pathways miRNA-34a-5p and miR-146a-5p could regulate in case of altered TREM2 expression. 

A7: Thanks for your suggestion, in the reviewed text we have include the new figure in supplementary data (Figure S3); the Figure S3 include a published data in order to  represent the possible pathways involved miRNA-34a-5p and miR-146a-5p to regulate TREM2 expression and /or inflammation.  

Round 2

Reviewer 3 Report

Manuscript is acceptable now